# Is Prehospital Assessment of qSOFA Parameters Associated with Earlier Targeted Sepsis Therapy? A Retrospective Cohort Study

**DOI:** 10.3390/jcm11123501

**Published:** 2022-06-17

**Authors:** André Dankert, Jochen Kraxner, Philipp Breitfeld, Clemens Bopp, Malte Issleib, Christoph Doehn, Janina Bathe, Linda Krause, Christian Zöllner, Martin Petzoldt

**Affiliations:** 1Department of Anesthesiology, Center of Anesthesiology and Intensive Care Medicine, University Medical Center Hamburg-Eppendorf, Martinistrasse 52, 20251 Hamburg, Germany; jochen-markmann@gmx.net (J.K.); p.breitfeld@uke.de (P.B.); m.issleib@uke.de (M.I.); christophdoehn@gmail.com (C.D.); j.bathe@uke.de (J.B.); c.zoellner@uke.de (C.Z.); m.petzoldt@uke.de (M.P.); 2Department of Anesthesiology and Intensive Care Medicine, German Military Hospital Hamburg, Lesserstrasse 180, 22049 Hamburg, Germany; clemensbopp@bundeswehr.org; 3Institute of Medical Biometry and Epidemiology, University Medical Center Hamburg-Eppendorf, Martinistrasse 52, 20251 Hamburg, Germany; l.krause@uke.de

**Keywords:** sepsis, systemic inflammatory response syndrome, organ dysfunction scores, emergency medicine, anti-infective agents, fluid therapy

## Abstract

Background: This study aimed to determine whether prehospital qSOFA (quick sequential organ failure assessment) assessment was associated with a shortened ‘time to antibiotics’ and ‘time to intravenous fluid resuscitation’ compared with standard assessment. Methods: This retrospective study included patients who were referred to our Emergency Department between 2014 and 2018 by emergency medical services, in whom sepsis was diagnosed during hospitalization. Two multivariable regression models were fitted, with and without qSOFA parameters, for ‘time to antibiotics’ (primary endpoint) and ‘time to intravenous fluid resuscitation’. Results: In total, 702 patients were included. Multiple linear regression analysis showed that antibiotics and intravenous fluids were initiated earlier if infections were suspected and emergency medical services involved emergency physicians. A heart rate above 90/min was associated with a shortened time to antibiotics. If qSOFA parameters were added to the models, a respiratory rate ≥ 22/min and altered mentation were independent predictors for earlier antibiotics. A systolic blood pressure ≤ 100 mmHg and altered mentation were independent predictors for earlier fluids. When qSOFA parameters were added, the explained variability of the model increased by 24% and 38%, respectively (adjusted R² 0.106 versus 0.131 for antibiotics and 0.117 versus 0.162 for fluids). Conclusion: Prehospital assessment of qSOFA parameters was associated with a shortened time to a targeted sepsis therapy.

## 1. Introduction

Sepsis is an acute life-threatening condition associated with a high incidence of septic shock, organ dysfunction and failure [1,2]. Despite advances in diagnostics and treatment, sepsis is still associated with a high mortality rate of approximately 10–47% [3,4,5,6,7]. Early recognition in the initial hours of sepsis is critically important if it translates into timely treatment, especially early administration of antibiotics and fluids, which in turn has a proven benefit for outcome and mortality [3,8,9].

Most frequently the first medical contact with patients suffering from sepsis is before hospitalization via emergency medical services (EMSs) [10]. However, at this stage, clinical signs are likely to be overlooked, underestimated or misinterpreted. Various screening tools have been used for the detection of sepsis, such as the systemic inflammatory response syndrome (SIRS) criteria or the sequential organ failure assessment (SOFA) score [3,5,10,11]; however, both scores rely on laboratory parameters from patient’s blood samples and are often not available in prehospital emergency medicine settings.

The quick SOFA (qSOFA) has been highlighted in the Third International Consensus Conference on the Definitions of Sepsis in 2016 and uses only three simple parameters: altered mentation, systolic blood pressure ≤ 100 mmHg and respiratory rate ≥ 22/min [5]. The qSOFA score is designed to promote early identification of sepsis [3], especially if more sophisticated diagnostic measures such as blood tests, biomarkers or imaging are unavailable, time-critical or results are pending. The qSOFA has been used in in-patient wards and emergency departments.

While the diagnostic performance of the qSOFA is limited, its universal availability and simple timely assessment based on routine clinical data make the qSOFA especially appealing for prehospital emergency medicine. Previous retrospective cohort studies found an association between the prehospital qSOFA score and in-hospital mortality [12,13]. However, the current role of qSOFA in prehospital emergency medicine is unclear. Does the assessment of qSOFA improve prehospital sepsis screening, allowing for earlier sepsis diagnoses and, most importantly, shorten the time to a targeted sepsis therapy?

This study aimed to determine whether prehospital assessment of qSOFA parameters was associated with a shortened ‘time to antibiotics’ (primary endpoint) and ‘time to fluid resuscitation’ compared with standard assessment of sepsis indicators.

## 2. Materials and Methods

This single-center retrospective cohort study was conducted in accordance with the Declaration of Helsinki. The Ethics Committee of the Medical Association of Hamburg waived the need to obtain informed consent for the collection and analysis of data (WF-041/18, 16 July 2018, chairman Prof. Dr. R. Stahl). The design and reporting were adapted to the Strengthening the Reporting of Observational Studies in Epidemiology (STROBE) guideline (Appendix A).

### 2.1. Data Acquisition

We used an anonymized data set, which was collected for internal quality assessment, for our statistical analysis. This anonymized data set encompasses the clinical and procedural data of adult patients with confirmed sepsis (discharge diagnosis) who were treated at the University Medical Center Hamburg-Eppendorf between 1 June 2014 and 30 June 2018. The inclusion criteria were minimum age 18 years, admission to the emergency department of the University Medical Center Hamburg-Eppendorf by an EMS and discharge diagnosis ‘sepsis’ (documented as an International Statistical Classification of Diseases and Related Health Problems, ICD-10). The exclusion criteria were patients who were secondarily transferred from another hospital, outpatients, patients with a clearly documented palliative care concept at the time of admission, admission due to primary trauma, pregnancy and unavailable or illegible emergency medical protocols. To avoid inclusion of hospital-acquired cases, patients were only included if the diagnosis of ‘sepsis’ was made within the first 48 h after admission. The German EMS system typically relies on two different levels of care, first a paramedic EMS (‘ambulance’) and second a physician-based EMS (paramedic and an emergency physician). In order to provide a high degree of flexibility, both units are used in a ‘rendezvous system’ [14,15].

### 2.2. Study Endpoints

The primary endpoint ‘time to antibiotics’ was defined as the time from the first medical contact to the first antibiotic dose documented in the patient’s chart. Secondary endpoints were ‘time to fluid resuscitation’ (from the first medical contact to the initiation of a targeted intravenous fluid therapy, defined as administration of more than 500 mL intravenous fluids during the initial therapy [10] (prehospital and/or emergency department), indicated by the documented administration of a second 500 mL dosage), intensive care admission rate, length of ICU and in-hospital stay and in-hospital mortality. The first medical contact was defined as the first documented prehospital contact of any member of the EMS with the patient.

### 2.3. Covariables

Potentially eligible covariables were identified through literature research, previous studies [3,5,10,16,17,18,19,20] and clinical considerations. Nine variables were considered:

Sociodemographic data
Age (years).Gender (male/female).

EMS
EMS without emergency physician (only paramedics) versus EMS with emergency physician (of note, this variable is specific to the German emergency system, which provides these two modular care levels).

Standard assessment bundle (sepsis indicators)
Suspected infection (yes/no), positive if the EMS clearly documented that any kind of infection was suspected.Abnormal body temperature (either <36 °C or >38 °C) (yes/no) [21].Heart rate ≥ 90/min (yes/no) [21].

qSOFA parameters
Respiratory rate ≥ 22/min (yes/no) [7].Systolic blood pressure ≤ 100 mmHg (yes/no) [7].Altered mentation (yes/no) [7].

If variables were absent from the protocol, we assumed they were unsuspicious.

### 2.4. Sample Size

This study is a pilot study with an explorative character (hypothesis generating); hence, data for a priori sample size calculation were unavailable. The number of available cases was restricted by the number of in-patients who were hospitalized with community-onset sepsis during the study period. To check whether our sample size was large enough to calculate multivariable regression models, we applied the approach proposed by Riley et al., which consists of four criteria [22]. The minimum sample size required to develop a prediction model for a continuous outcome (e.g., ‘time to antibiotics’) with a mean of 238 and a standard deviation of 195, containing nine predictors and an anticipated R^2^ of 0.10, was 579. For the secondary endpoint, ‘time to fluid resuscitation’, with a mean of 262 and a standard deviation of 202, the same sample size was calculated. We concluded that our sample size of 702 is sufficiently large to calculate the planned multivariable regression models.

### 2.5. Descriptive Statistics

Sample characteristics are given as absolute and relative frequencies, median (IQR) or mean (standard deviation). All time variables were skewed and, therefore, logarithmized. For comparative analysis Student’s *t*-test, Mann–Whitney U-Test or Fisher’s exact test were used, whichever was appropriate. We use *p*-values only as descriptive summary measures and do not regard them as results of confirmatory testing; thus, we report nominal *p*-values without correction for multiplicity. For multiple linear regression analysis, standardized coefficients with 95% confidence interval (95% CI) are reported. Statistical analysis was performed using SPSS version 25.0 (IBM, Armonk, NY, USA).

### 2.6. Multiple Linear Regression Models

To answer our primary and secondary research questions, four multiple linear regression models were developed that solely rely on prehospital assessed parameters. Complete case analysis was used for model fitting. We used the enter method as provided by SPSS. Nine available, potentially eligible covariables were used for the fitting of the multiple linear regression models without any additional variable selection steps (forced variables) as both the prehospital sepsis indicators (standard parameters) and qSOFA parameters were considered given assessment bundles, prespecified by the pertinent literature [10,19,23,24,25].

The first multiple linear regression model was fitted to assess the association with the primary endpoint ‘time to antibiotics’ without considering qSOFA parameters, while all other predetermined covariables were included (model without qSOFA). In the second model, qSOFA parameters were added, as it was intended to evaluate the incremental value of the qSOFA parameters on top of the clinical standard (model with qSOFA). The third model was fitted to assess the association with the secondary endpoint ‘time to fluid resuscitation’ without using qSOFA parameters, while the fourth model, again, added all qSOFA parameters. The adjusted coefficient of determination (R^2^), which describes the percentage of variance of the outcome explained by the predictors, was calculated to assess whether the additional qSOFA parameters add value to the overall model and improve the goodness of the model fit.

## 3. Results

### 3.1. Data Acquisition and Cohort Characteristics

We identified 4214 in-patients with confirmed sepsis, who were treated at the University Medical Center Hamburg-Eppendorf between 1 June 2014 and 30 June 2018. In total, 702 patients fulfilled all eligibility criteria and were included in the analysis (Figure 1). Prehospital emergency medical care was provided by EMS without physicians in 64.2% (451/702) or by EMS with emergency physicians in 35.8% (251/702) of the cases (Table 1).

EMS with emergency physicians more frequently treated patients with severe sepsis (2.7% versus 4.0%; discharge diagnosis), septic shock (9.5% versus 20.7%; discharge diagnosis), respiratory focus (27.9% versus 50.6%) and altered mentation (24.2% versus 44.6%, *p* < 0.001); however, we did not observe a difference in either the length of in-hospital stay (11 days [IQR 7–20 days] versus 13 days [IQR 6–22 days], *p* = 0.814) or in-hospital mortality (10.6% versus 11.2%, *p* = 0.834) between both groups. 

### 3.2. Assessment of Sepsis Indicators

Overall, the prehospital assessment of sepsis indicators and qSOFA parameters was poor, particularly regarding suspected infections, body temperature and respiratory rate. EMS without emergency physicians less frequently provided complete recordings of all five focused vital parameters compared with EMS with emergency physicians (8.6% versus 38.6%, *p* < 0.001) (Figure 2).

Patients with a positive qSOFA score (two or more criteria) received antibiotics (median 118.0 min [IQR 86.0–186.0 min] versus median 194.5 min [IQR 127.5–302.0 min], *p* < 0.001) and intravenous fluid resuscitation (median 104.0 min [IQR 70.3–143.5 min] versus median 183.0 min [IQR 103.0–346.0 min], *p* < 0.001) earlier than those without a positive qSOFA score.

### 3.3. Multiple Linear Regression Models

All multiple linear regression models were adjusted for age, gender and type of EMS, as the team structure of the EMS was independently associated with a reduced ‘time to antibiotics’ and ‘time to fluid resuscitation’ (Figure 3 and Figure 4).

Multiple linear regression analysis revealed that clinically suspected infection (standardized coefficient of the logarithmized time β −0.138, 95% CI −0.228 to −0.047) and heart rate ≥90/min (β −0.113, 95% CI −0.193 to −0.032) were associated with a shortened ‘time to antibiotics’ (model without qSOFA, Figure 3). If qSOFA parameters were added to the linear regression model, a respiratory rate ≥22/min (β −0.144, 95% CI −0.226 to −0.062) and altered mentation (β −0.097, 95% CI −0.182 to −0.012) were independently associated with a shortened ‘time to antibiotics’ (Figure 3), while clinically suspected infection (β −0.099, 95% CI −0.190 to −0.008), a systolic blood pressure ≤100 mmHg (β −0.085, 95% CI −0.165 to −0.005) and altered mentation (β −0.214, 95% CI −0.300 to −0.128) were independently associated with a shortened ‘time to fluid resuscitation’ (Figure 4).

Our data revealed an incremental value of qSOFA parameters for both linear regression models. When qSOFA parameters were added to the model, the explained variability increased by 24% and 38%, respectively (adjusted R² 0.106 versus 0.131 for antibiotics and adjusted R² 0.117 versus 0.162 for fluids).

## 4. Discussion

The prehospital suspicion of ‘sepsis’ relies on the distinct interpretation of multiple clinical signs. Prehospital EMSs play an important role in the early detection of typical sepsis signs and could be key drivers for an early targeted therapy, which in turn could prevent progression of sepsis, severe sepsis and septic shock. Our data indicate that this great potential has not yet been effectively exploited. Sepsis is still underdiagnosed in prehospital emergency medicine [24]. Our findings suggest that the quality of prehospital assessment of sepsis indicators is still poor, particularly regarding the recording of suspected infections, body temperature and respiratory rate. The quality of the recording increased if an emergency physician joined the EMS. Nevertheless, even if adjusted for the EMS team structure, the assessment of sepsis indicators and qSOFA parameters was strongly associated with faster targeted therapy, empiric antibiotic treatment and fluid resuscitation. Here, qSOFA parameters had an incremental value on top of a standard assessment.

The 2021 Surviving Sepsis Campaign guidelines recommend giving antibiotics within three hours after the first recognition of possible sepsis and within one hour in individuals with a high likelihood of sepsis or possible septic shock [3]. Crystalloid fluids should be given within the first three hours [3]. Our data indicate that early recognition of sepsis indicators by prehospital emergency medical services might contribute to reaching these goals.

Even though antibiotic application in the ambulance by specially trained EMS personnel did not lead to improved survival in a multicenter trial [26], a recent publication indicated that each additional hour of door-to-antibiotic time in the emergency department was associated with 10% increased odds of 1-year mortality [27]. A previous study showed, that out-of-hospital care may streamline the in-hospital processes of patients with suspected sepsis, by shortening the ‘time to antibiotics’ and ‘time to intravenous fluids’ in the emergency department [28]. Here, our data suggest, that the time from the onset of sepsis to antibiotic administration can be reduced if EMSs quickly recognize and clearly notify suspected sepsis, label these patients as high priority and promptly transfer them to an appropriate destination (e.g., emergency room).

Our data show that, beyond suspected infection and standard recordings of body temperature and heart rate, assessment of qSOFA criteria had an additional value in prehospital emergency medicine. However, in our opinion, prehospital sepsis screening should be systematic, protocol-based, coupled with a clearly outlined workflow and complemented with structured training programs and education campaigns.

Previous retrospective cohort studies have demonstrated the association between the prehospital qSOFA score and in-hospital mortality [12,13]. However, most studies investigated the value of the clinical qSOFA to predict mortality [25,29,30]. As the qSOFA is a very simple score, its diagnostic accuracy is limited. Various meta-analysis found low sensitivity of the qSOFA in conjunction with a moderate specificity in patients with suspected sepsis in the ICU, emergency department or hospital ward [25,29,30]; this in turn limits its applicability as a screening tool. Current guidelines do not recommend qSOFA as single screening tool for sepsis or septic shock [3], and it has been proposed that a combination of qSOFA and SIRS criteria might be favorable [25]. Our findings indicate that, regarding the flow and timing of a targeted sepsis therapy, qSOFA might be beneficial and should be used on top of a standard clinical assessment. The universal availability and simple timely assessment of routine data makes the qSOFA appealing for prehospital emergency medicine. However, data in prehospital emergency medical settings are still very sparse [16,31,32].

The retrospective nature of this study is a well-recognized limitation and a potential source for selection bias. Sepsis screening by means of qSOFA parameters can only indirectly impact the time to treatment by increasing the detection rate of sepsis and improving severity grading in prehospital emergency settings. Our analysis only outlines associations between variables but not causality. As EMS systems relevantly differ between countries and regions, we decided to adjust our multiple regression model for the ‘type of EMS’ as we believe that this might strengthen the generalizability and applicability of our findings. Our data were gathered in only one university hospital; since standards and infrastructural prerequisites differ among regions, our findings cannot be generalized or extrapolated to other healthcare systems or populations without critical appraisal.

## 5. Conclusions

Our data suggest that prehospital screening for sepsis signs by the EMS is associated with an earlier initiation of a targeted sepsis therapy and shortened time to empirical antibiotic therapy and intravenous fluid resuscitation. Assessment of the qSOFA parameter had an incremental value on top of a standard assessment.

## Figures and Tables

**Figure 1 jcm-11-03501-f001:**
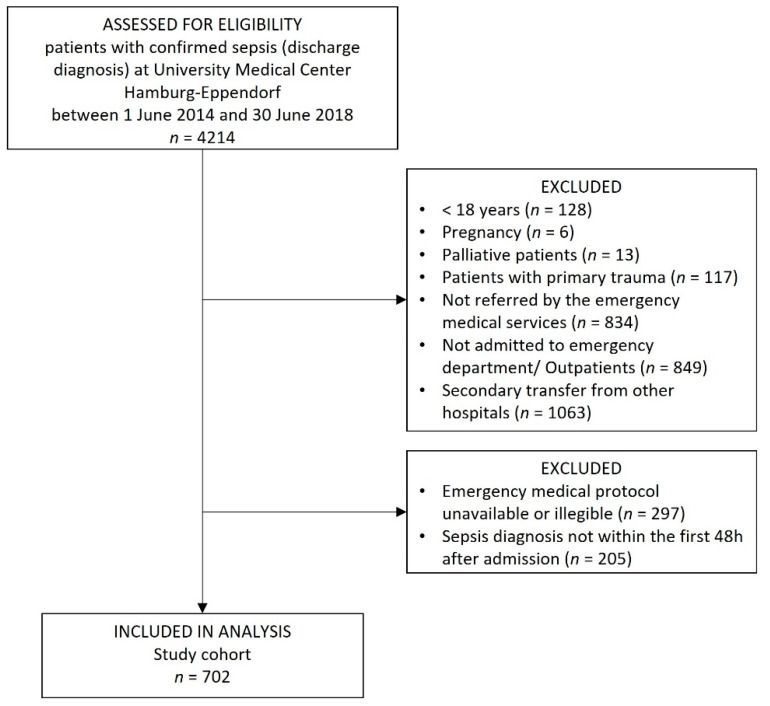
Study flow.

**Figure 2 jcm-11-03501-f002:**
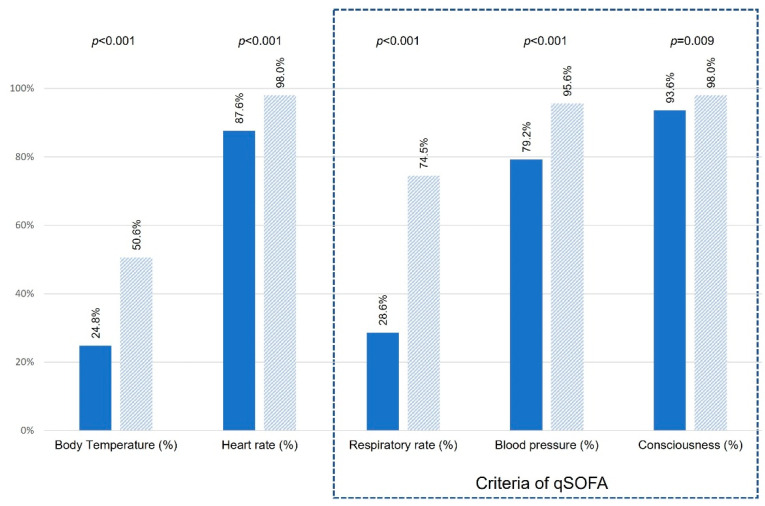
Frequency (%) of prehospital recorded sepsis indicators (standard assessment) and qSOFA parameters subdivided in two groups: EMS paramedics only (dark blue) and EMS with emergency physicians (light blue), *p*-values were calculated between both groups using Fisher’s exact tests; qSOFA parameters are highlighted in a box (right, blue grid lines).

**Figure 3 jcm-11-03501-f003:**
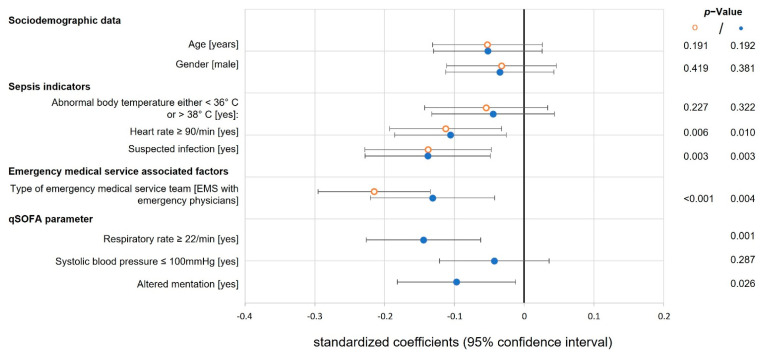
Time to antibiotics. The forest plot shows the standardized coefficients with 95% CI (x-axis) and *p*-values for each covariable regarding the primary endpoint ‘time to antibiotics’ of two multiple linear regression models: model without qSOFA (orange ᵒ dot) and model with additional qSOFA parameters (blue • dot); time is logarithmized; the vertical solid line marks a standardized coefficient of zero, a shift to the right indicates a prolonged and to the left side a shortened ‘time to antibiotics’; qSOFA: quick sequential organ failure assessment.

**Figure 4 jcm-11-03501-f004:**
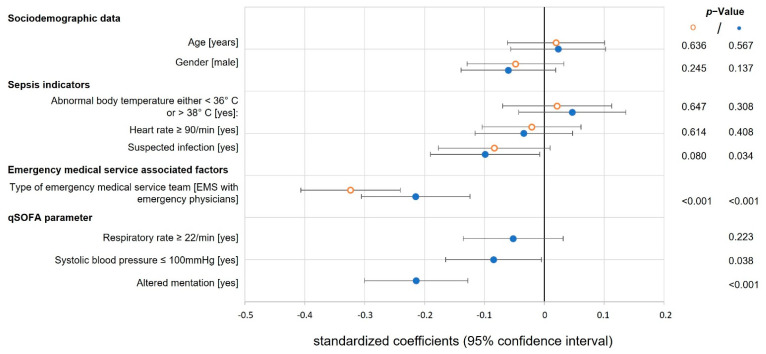
Time to fluid resuscitation. The forest plot shows the standardized coefficients with 95% CI (x-axis) and *p*-values for each covariable regarding the secondary endpoint ‘time to fluid resuscitation’ of two multiple linear regression models (*n* = 702 cases): model without qSOFA (orange ᵒ dot) and model with additional qSOFA parameters (blue • dot); time is logarithmized; the vertical solid line marks a standard coefficient of zero, a shift to the right indicates a prolonged and to the left side a shortened ‘time to fluid resuscitation’; qSOFA: quick sequential organ failure assessment.

**Table 1 jcm-11-03501-t001:** Baseline characteristics, disease characteristics, sepsis indicators and outcome of the study cohort.

	All Patients
	(*n* = 702)
Sociodemographic data	
Age (years)	68.4 ± 14.9
Gender male	411 (58.5%)
Emergency service data	
Urban area	645 (91.9%)
Admission to emergency room	197 (28.1%)
EMS with paramedics only	451 (64.2%)
EMS with emergency physicians	251 (35.8%)
Final diagnosis (at discharge)	
Sepsis	585 (83.3%)
Severe sepsis	22 (3.1%)
Septic shock	95 (13.5%)
Sepsis focus (discharge diagnosis)	
Respiratory	253 (36.0%)
Genitourinary tract	180 (25.6%)
Abdominal	96 (13.7%)
Blood stream	54 (7.7%)
Skin	36 (5.1%)
CNS	12 (1.7%)
Other	29 (4.1%)
Unknow	42 (6.0%)
Prehospital assessed sepsis indicators ^b^	
Abnormal body temperature ^a^	165 (23.5%)
Elevated heart rate (>90/min) ^a^	468 (66.6%)
Suspected infection ^a^	242 (34.5%)
Prehospital assessed qSOFA parameters ^b^	
Respiratory rate ≥ 22/min ^a^	80 (11.4%)
Systolic blood pressure ≤ 100 mmHg ^a^	241 (34.4%)
Altered mentation ^a^	223 (31.7%)
Primary and secondary endpoints	
Time to antibiotics (min) ^c^	238.1 ± 195.2
Time to fluid resuscitation (min) ^d^	255.3 ± 254.3
ICU admission rate	535 (76.2%)
Length of ICU stay (days)	7 (4–14)
Length of hospital stay (days)	11 (7–21)
In-hospital mortality (%)	76 (10.8%)

Values reported as *n* (%), mean (±SD) or median (IQR). ^a^ If variables were absent from the protocols, we assume they were unsuspicious; ^b^ assessed at the time of first medical contact; ^c^ defined as time from the first medical contact (paramedic and/or emergency physician) to the beginning of the antibiotic administration; ^d^ defined as time from the first medical contact (paramedic and/or emergency physician) to the beginning of fluid resuscitation; abnormal body temperature was defined as either <36 °C or >38 °C; abbreviation: CNS: central nervous system; EMS: emergency medical service (paramedic with or without physician); ICU: intensive care unit; qSOFA: quick sequential organ failure assessment; SD: standard deviation.

## Data Availability

Not applicable.

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
