# Peer review of "Is Prehospital Assessment of qSOFA Parameters Associated with Earlier Targeted Sepsis Therapy? A Retrospective Cohort Study"

_jcm, 2022, doi:10.3390/jcm11123501_

Round 1

Reviewer 1 Report

The authors report the findings of a single-center- retrospective cohort study on adult patients, who were referred to the Emergency Department of an academic center by emergency medical services (EMS) over a 4-year period, and who were diagnosed with sepsis within 48 hours after admission. Prehospital assessment of qSOFA assessment was associated with a decreased time to initiation of antibiotics and time to intravenous fluid resuscitation compared to standard assessment of sepsis indicators.

This is a well-written manuscript that underscores the importance of a systematic prehospital sepsis screening in timely initiation of antibiotics. Although similar studies have been published recently (eg

- Shu et al. Pre-hospital qSOFA as a predictor of sepsis and mortality. AJEM 2019, 37(7), 1273-8

- Koyama et al. Use of prehospital qSOFA in predicting in-hospital mortality in patients with suspected infection: A retrospective cohort study. PLoS One 2019. doi:10.1371/journal.pone.0216560),  all these studies were in systems with paramedic only EMS. Please provide more information about the EMS Emergency physician care level.

Clarify when infection was suspected. Table 1 under prehospital assessed sepsis indicators lists for suspected infection “Defined as time from the first medical contact (paramedic and/or emergency physician) to the begin of the antibiotic administration”  

Reviewer 2 Report

In this work, the authors evaluate the impact of collecting qSOFA parameters on time to treatment in sepsis patients. It is an interesting article since the vast majority of works to date have been based on studying the usefulness of qSOFA as an identifier of severity and predictor of mortality. I consider this is an interesting article to remind us of the benefits of a score as easy and simple to measure as the qSOFA, some points should be clarified.

-        The authors say compared the determination of qSOFA against the "standard assessment". What is the standard assessment? According to the parameters included as co-variables those would be: age, gender, suspicion of infection, temperature, and heart rate. With so few variables evaluated, it is quite feasible that the inclusion of new variables, in this case those of qSOFA, will improve the regression models. There are no other clinical parameters recorded (comorbidities, immunosuppression, or chronic treatments,…)?

In line, What are the clinical features/symptoms that EMS personnel use to define suspected infection?

-        I have not understood what is the objective of separating patients into those with and without a physician. As I understand it, it is for noting that those who have had a physician have recorded a higher percentage of qSOFA data. However, some of the conclusions from dividing the cohort in this way should be made with caution. It seems clear following the baseline characteristics of Table 1 that patients who have been managed in EMS with a physician are more seriously ill. For this reason, it is logical that more information is recorded in these more severe patients and they also receive antibiotic treatment before less severe patients. In fact, it can be seen that mortality is even slightly higher in patients with a physician despite a better time to treatment. Therefore, the discussion of the results at the point of comparing with and without a physician should be expanded.

Minor points:

-        The supplementary file includes the unfilled strobe checklist, it appears as it is downloaded from the web. Please upload the completed version with the information from your study.

-        What is the used regression model method? Enter, Wald, forward,… I'm guessing it's enter method, since it says it's not by steps, but please clarify.

-        Table 1 legend: It says “values reported as n (%)”. How ever there is no n in the table only %.

Reviewer 3 Report

Authors tried to investigate the association between prehospital assessment of qSOFA and earlier targeted sepsis therapy, and concluded that prehospital screening for sepsis signs by the EMS is associated with an earlier initiation of a targeted sepsis therapy manifested by shortened time to antibiotics and fluid resuscitation.

It is important issue, but the methods they used is not sufficient to solve this issue.

They used multiple linear regression models to investigate the association btw assessment of qSOFA and earlier treatment. However, the findings they showed is association btw prehospital severity of sepsis and earlier treatment, not association btw assessment/screening of qSOFA and earlier treatment. It is totally different. The more severe sepsis in prehospital/ED is, the earlier treatment might be applied. For example, septic shock patients received early antibiotics more than sepsis patients (PMID: 35027073).

In Table 1, they showed baseline characteristics and outcomes of sepsis according to the two subgroups, i.e. with of without emergency physician. I am not sure what these comparisons implies in this study.

Round 2

Reviewer 2 Report

The authors have properly answered all my questions. With the current modifications, the message of the manuscript is understood in a easier way.

Reviewer 3 Report

None